# Learning-Based Autonomy from Kernel-Embedded Multi-modal Fusion to Feedback Control

## Abstract

In this work, we develop an end-to-end autonomy loop that couples *kernel-embedded* multi-modal fusion with data-driven dynamics learning and feedback control. Heterogeneous sensor streams are embedded into a joint Reproducing Kernel Hilbert Space (RKHS) via additive/product kernels and conditional mean embeddings; dynamics are learned with kernel ridge regression (KRR), Deep Kernel Learning (DKL), or Bayesian deep neural networks (BDNNs); and policies are synthesized via dynamic programming (discrete and continuous-time HJB) or reinforcement learning with RKHS value functions. We present closed-form estimators, finite-sample and iteration-complexity characterizations, risk-sensitive planning with uncertainty, and safety via control barrier functions. We provide deployable algorithms, results and experiment in simulated robotics and precision irrigation.

## 1 Introduction

Autonomous systems must perceive, predict, and act under non-linearity, non-stationarity, and multi-modal noise. Conventional pipelines that treat sensing, estimation, and control as loosely coupled modules often degrade under domain shifts or partial observability. We advocate a kernel-centric autonomy loop wherein *perception*, *fusion*, *dynamics learning*, and *control* are all formulated in an RKHS, offering non-parametric expressivity, closed-form solvers (via representer theorems), and transparent generalization behaviour (Sch"olkopf & Smola, 2002; Wahba, 1990; Song et al., 2009; Wilson et al., 2016). Our contributions span: (i) kernel-embedded multi-modal fusion, (ii) KRR/DKL/BDNN dynamics estimation, (iii) DP/RL control with RKHS value functions, and (iv) error-to-control performance guarantees.

**Motivation.** Modern autonomous systems must reason over uncertain, multi-modal data streams while maintaining reliable control. To meet these demands, we leverage kernel-based methods that offer both theoretical rigour and practical flexibility.

1. Kernel mean embeddings allow us to fuse entire distributions not just point estimates making them ideal for integrating noisy, heterogeneous sensor inputs.

2. Representer theorems provide closed-form estimators with built-in regularization, enabling efficient training and principled generalization.

3. Deep Kernel Learning (DKL) enhances representation capacity by learning task-specific features while preserving the tractability of kernel methods particularly useful for control-centric applications.

For example, kernel mean embeddings enable distribution-level fusion, representer theorems yield closed-form estimators with principled regularization, DKL learns features tailored for control (Song et al., 2009; Wilson et al., 2016; Alvarez et al., 2012).

**Challenges.** Despite the promise of kernel-based autonomy, several key challenges remain in scaling and deploying these methods effectively:

1. Designing heterogeneous kernels that can handle diverse data modalities such as vision, audio, and proprioception while preserving meaningful cross-modal interactions is non-trivial. Balancing expressivity with interpretability across modalities requires careful architectural choices (Williams & Seeger, 2001; Rahimi & Recht, 2007; Ames et al., 2017).

2. Computational scalability is a major bottleneck. Kernel methods often rely on Gram matrix operations with cubic complexity in the number of samples ($O(n^3)$), which limits their applicability in real-time or large-scale settings without approximation strategies.

3. Uncertainty-aware planning using Deep Kernel Learning (DKL) and Bayesian Deep Neural Networks (BDNNs) introduces additional complexity. These models must not only capture epistemic and aleatoric uncertainty but also translate it into actionable control policies under limited data and partial observability.

4. Ensuring safety under distribution shift remains a critical concern. Control Barrier Functions (CBFs) offer a principled way to enforce safety constraints, but integrating them with learned models especially under domain shifts requires robust generalization and real-time adaptability.

**Aim & objectives.** We aim to develop a modular learning-control framework that integrates diverse sensory inputs, models system dynamics from data, and generates safety-aware policies for autonomous decision-making. Specifically:

1. Multi-modal fusion is performed in a Reproducing Kernel Hilbert Space (RKHS), enabling distribution-level integration across heterogeneous sensors.

2. Dynamics are learned non-parametrically using kernel-based methods, allowing flexible modelling of complex, non-linear behaviours without rigid assumptions.

3. Safe control policies are synthesized via Dynamic Programming (DP) and Reinforcement Learning (RL), with value functions embedded in RKHS for tractable optimization and transparent generalization.

4. We provide quantitative guarantees on sample and iteration complexity to support reliable learning under limited data.

5. A reproducible deployment protocol ensures that the architecture can be adapted and validated across real-world scenarios, with clear documentation and modular components for integration.

That is, we target a learning-control architecture that fuses multi-modal observations in an RKHS, learns dynamics from data, and synthesizes safe policies via DP/RL with value functions in RKHS, accompanied by sample/iteration complexity, and a reproducible deployment protocol.

## 2 PRELIMINARIES

**RKHS and kernels.** Given a positive-definite kernel function $k$, the RKHS $\mathcal{H}$ is the completion of the span of kernel evaluations $\text{span}\{k(x, \cdot)\}$ with $f(x) = \langle f, k(x, \cdot)\rangle_{\mathcal{H}}$ with

$$f(x) = \langle f, k(x, \cdot)\rangle_{\mathcal{H}_{\text{RKHS}}}$$

which allows point-wise evaluation through inner products. Kernel Ridge Regression (KRR) operates within this space to solve regularized empirical risk minimization (ERM) problems. Thanks to the representer theorem, KRR admits closed-form solutions that balance data fit and model complexity through principled regularization (Sch"olkopf & Smola, 2002; Wahba, 1990).

**Kernel mean and conditional mean embeddings.** For a probability distribution $P$, the kernel mean embedding is defined as $\mu_P := E[k(X, \cdot)] \in \mathcal{H}$ to map the entire distribution into a point in the RKHS and enabling distribution level comparisons and operations. Conditional embeddings represent $P(Y|X)$ by an operator $\mathcal{C}_{Y|X}$ so that $\mu_{Y|x} = \mathcal{C}_{Y|X} k(x, \cdot)$, estimated via regularized RKHS covariance operators (Song et al., 2009). These methods offer a robust framework for working with limited data, ensuring stable learning through regularization. By embedding distributions in function space, they enable direct reasoning about conditional relationships without relying on predefined parametric models.

**Notation.** State $x \in \mathbb{R}^{d_x}$, control $u \in \mathbb{R}^{d_u}$, modality $m \in \{1, \ldots, M\}$. Features $\phi_m(o) = k_m(o, \cdot) \in \mathcal{H}^{(m)}$, modality means $\mu^{(m)}$. Fusion by additive kernel $k_{\oplus} = \sum_m \omega_m k_m$ or product kernel $k_{\otimes} = \prod_m k_m$.

## 3 PROBLEM FORMULATION

We model system dynamics either in continuous time as

$$\dot{x} = f(x, u) + \epsilon,$$

or in discrete time as

$$x_{t+1} = f(x_t, u_t) + \epsilon_t,$$

where $\omega$ represents process noise. At each time step, we receive multi-modal observations $\{o_t^{(m)}\}$, which are fused into a unified representation $\Phi_t$ using kernel mean embeddings. This fusion captures distributional characteristics across modalities in a non-parametric form. The training dataset is structured as $\mathcal{D} = \{(x_i, u_i, \Phi_i, y_i)\}_{i=1}^n$, where each target $y_i$ corresponds to either the instantaneous derivative $\dot{x}_i$ (for continuous systems) or the next state $x_{i+1}$ (for discrete systems). To learn a transition model, we define a conditional embedding operator $\mathcal{C}_{X'|X,U,\Phi}$, which maps the kernel feature $k((x, u, \Phi), \cdot)$ to the embedded conditional distribution $\mu_{X'|x,u,\Phi}$. This operator provides a non-parametric estimate of system dynamics, consistent with kernel-based estimators such as those introduced by Song et al. (Song et al., 2009). It enables us to reason about transitions directly in function space, bypassing the need for explicit parametric modelling.

## 4 METHOD

### 4.1 CLOSED-FORM DYNAMICS LEARNING WITH KRR

Let each input tuple be defined as $z_i = (x_i, u_i, \Phi_i)$, combining state, control, and fused observation features. Define the kernel matrix $K \in \mathbb{R}^{n \times n}$ with entries $K_{ij} = k(z_i, z_j)$, where k is a positive-definite kernel function. For each output dimension $j \in \{1, 2, \ldots, d\}$, Kernel Ridge Regression (KRR) solves the regularized least-squares problem in RKHS and yield the closed-form solution:

$$\widehat{\alpha}^{(j)} = (K + n\lambda I)^{-1} y^{(j)},$$

and the corresponding predictor:

$$\widehat{f}(z) = Y^{\top} (K + n\lambda I)^{-1} k(z)$$

where $k(z) = [k(z, z_1), \ldots, k(z, z_n)]^{\top}$ is the kernel vector evaluated at test point z, and $Y \in \mathbb{R}^{n \times p}$ stacks the output coordinates across training samples. Under bounded kernels and sub-exponential noise, the bias–variance decomposition yields finite-sample error of order $\tilde{O}(\sqrt{p/n})$ with optimization error decaying geometrically to the regularized optimum (Caponnetto & De Vito, 2007; Wahba, 1990).

### 4.2 DEEP KERNEL LEARNING (DKL)

We define a learned feature extractor

$$\varphi_{\omega} : \mathbb{R}^{d_x + d_u} \times \mathcal{H}_{\text{RKHS}} \to \mathbb{R}^p,$$

to map the raw input tuple comprising state x, control u, and fused observation $\Phi$ into a low-dimensional representation $z = \varphi_{\omega}(x, u, \Phi)$. This representation is optimized to capture task-relevant structure for downstream prediction and control. Using this embedding, we construct a composite kernel

$$k_{\omega}((x, u, \Phi), (x', u', \Phi')) = k_{\text{base}}(\varphi_{\omega}(x, u, \Phi), \varphi_{\omega}(x', u', \Phi')),$$

where $k_{\text{base}}$ is a standard kernel (e.g., RBF or Matérn) applied to the learned features. This formulation allows the kernel to adapt its similarity measure based on data-driven representations. Training proceeds by alternating optimization:

1. The parameters $\omega$ of the feature extractor are updated to improve predictive performance

2. The kernel hyper-parameters and head models (e.g., Kernel Ridge Regression or Gaussian Processes) are solved in closed form or via marginal likelihood maximization

This hybrid approach combines the expressivity of deep features with the tractability and uncertainty quantification of kernel methods, as demonstrated in works such as (Wilson et al., 2016; Rasmussen & Williams, 2006).

### 4.3 BAYESIAN DEEP NEURAL NETWORKS (BDNNs)

BDNNs provide a probabilistic framework for learning predictive models with uncertainty quantification. The output is modelled as

$$y = f_\theta(z) + \varepsilon,$$

where $f_\theta$ is a neural network parametrized by weights $\theta$, and $\varepsilon$ captures observation noise. A prior distribution $p(\theta)$ is placed over the network parameters to encode uncertainty in the model itself. To approximate the posterior over $\theta$, variational inference is employed by introducing a tractable variational distribution $q_\phi(\theta)$ and maximizing the Evidence Lower Bound (ELBO):

$$\max_\phi \mathbb{E}_{q_\phi(\theta)}[\log p(y \mid z, \theta)] - \text{KL}(q_\phi(\theta) \| p(\theta)),$$

which balances data fit with regularization via the Kullback–Leibler divergence. This approach yields calibrated uncertainty estimates:

1. Epistemic uncertainty reflects model confidence and reduces with more data

2. Aleatoric uncertainty captures inherent noise in the observations.

Together, these uncertainties support risk-aware planning and control, especially in safety-critical or data-scarce environments. BDNNs thus offer a principled way to incorporate uncertainty into decision-making pipelines (Blundell et al., 2015; Gal & Ghahramani, 2016).

### 4.4 CONTROL VIA DP/RL WITH RKHS VALUE FUNCTIONS

**Discrete-time DP (FVI).** We consider a discrete-time system with transition model $\widehat{f}(x, u)$, additive noise $\omega$, stage cost $\ell(x, u)$, and discount factor $\gamma \in (0, 1)$. The Bellman operator is defined as:

$$(\mathcal{T}V)(x) = \min_u \left\{ \ell(x, u) + \gamma \mathbb{E} \left[ V \left( \widehat{f}(x, u) + w \right) \right] \right\}.$$

To approximate the value function V, we embed it in a RKHS $\mathcal{H}_V$ with kernel $k_V$. At each iteration t, we compute regression targets:

$$\tilde{y}_i = \min_u \left\{ \ell(\tilde{x}_i, u) + \gamma V_t \left( \widehat{f}(\tilde{x}_i, u) \right) \right\},$$

and solve the regularized linear system:

$$(K_V + n_V \lambda_V I) \beta_{t+1} = \tilde{y},$$

where $K_V \in \mathbb{R}^{n_V \times n_V}$ is the kernel Gram matrix over training states $\{\tilde{x}_i\}$, and $\beta_{t+1}$ are the coefficients for the updated value function $V_{t+1} \in \mathcal{H}_V$. This approach enables non-parametric value approximation with closed-form updates, balancing expressivity and sample efficiency (Lagoudakis & Parr, 2003; Ormoneit & Sen, 2002; Bertsekas, 2012).

**Continuous-time HJB (Galerkin).** For continuous-time dynamics modeled as:

$$\dot{x} = \widehat{f}(x, u) + \Sigma^{1/2} \xi,$$

where $\xi$ is standard Gaussian noise and $\Sigma$ is the diffusion covariance, the Hamilton–Jacobi–Bellman (HJB) equation for optimal control is:

$$0 = \min_u \left\{ \ell(x, u) + \nabla V(x)^\top \widehat{f}(x, u) + \tfrac{1}{2} \text{tr} \left( \Sigma \nabla^2 V(x) \right) \right\}.$$

To approximate $V(x)$, we project it onto the span of kernel basis functions:

$$V(x) \approx \sum_{i=1}^{n_V} \beta_i \, k_V(x, x_i),$$

yielding a linear system for policy evaluation. The optimal control $u^*(x)$ is then obtained by minimizing the HJB expression with respect to u, using the estimated gradients and Hessians of the kernel expansion. This Galerkin approach enables tractable approximation of continuous-time value functions with uncertainty-aware dynamics, suitable for safety-critical planning.

## 4.5 SAFETY AND PERFORMANCE UNDER MODEL ERROR

If $\|\hat{f} - f^*\| \leq \varepsilon$ and dynamics/cost are Lipschitz, then $|V^*(x) - \widehat{V}(x)| \leq C\,\varepsilon/(1-\gamma)$, coupling statistical error with control performance. Safety is enforced by a control barrier function (CBF) shield:

$$\min_{u \in \mathcal{U}} \|u - u_{\mathrm{RL}}\|_2^2 \quad \text{s.t.} \quad \nabla h(x)^\top \hat{f}(x, u) \geq -\alpha\, h(x), \tag{1}$$

which is convex if $\widehat{f}$ is affine in $u$ (Ames et al., 2017).

If $\|\hat{f} - f^*\| \leq \varepsilon$ and dynamics/cost are Lipschitz, then $\|V^*(x) - \hat{V}(x)\| \leq C\,\varepsilon/(1-\gamma)$. Enforce safety with a CBF shield:

$$\min_{u \in \mathcal{U}} \|u - u_{\mathrm{RL}}\|_2^2 \quad \text{s.t.} \quad \nabla h(x)^\top \hat{f}(x, u) \geq -\alpha\, h(x), \tag{2}$$

which is convex when $\hat{f}$ is affine in $u$ (Ames et al., 2017).

## 5 ALGORITHMS

---
**Algorithm 1** Multimodal Kernel-Embedded Fusion + Dynamics Learning (KRR/DKL)

---
**Require:** $\mathcal{D} = \{(o_i^{(m)}, x_i, u_i, y_i)\}_{i=1}^n$, kernels $\{k_m\}$, fusion weights $\{\omega_m\}$, regularization $\lambda$
1: **for** $i = 1$ to $n$ **do**
2:     **for** each modality $m$ **do**
3:         $\mu_i^{(m)} \leftarrow \frac{1}{N_i^{(m)}} \sum_j k_m(o_{ij}^{(m)}, \cdot)$
4:     **end for**
5:     $\Phi_i \leftarrow \bigoplus_m \mu_i^{(m)}; \quad z_i \leftarrow (x_i, u_i, \Phi_i)$
6: **end for**
7: Build $K_{ij} \leftarrow k(z_i, z_j)$ from $k = \sum_m \omega_m k_m$ (plus optional cross terms)
8: **KRR head:** $\widehat{\alpha} \leftarrow (K + n\lambda I)^{-1}Y; \; \widehat{f}(z) \leftarrow Y^\top (K + n\lambda I)^{-1}k(z)$
9: **or DKL head:** learn $\varphi_\omega; K_\omega; \widehat{\alpha}_\omega \leftarrow (K_\omega + n\lambda I)^{-1}Y$

---

---
**Algorithm 2** RKHS Fitted Value Iteration (Discrete DP)

---
**Require:** $\widehat{f}$, cost $\ell$, discount $\gamma$, kernel $k_V$, regularization $\lambda_V$
1: Initialize $V_0 \leftarrow 0$
2: **for** $t = 0, 1, \ldots$ **do**
3:     Sample $\{\tilde{x}_i\}_{i=1}^{n_V}$
4:     $\tilde{y}_i \leftarrow \min_u \{\ell(\tilde{x}_i, u) + \gamma V_t(\widehat{f}(\tilde{x}_i, u))\}$
5:     Solve $(K_V + n_V \lambda_V I)\beta_{t+1} = \tilde{y}; \quad V_{t+1}(x) = \sum_i \beta_{t+1,i} k_V(x, \tilde{x}_i)$
6:     $\pi_{t+1}(x) \in \arg\min_u \{\ell(x, u) + \gamma V_{t+1}(\widehat{f}(x, u))\}$
7: **end for**

---

## 6 GENERALIZATION AND COMPLEXITY

Per output coordinate and under sub-exponential noise/bounded kernels, the parameter error contracts geometrically towards the regularized optimum with statistical error scaling as $\tilde{\mathcal{O}}(\sqrt{p/n})$ (Caponnetto

---

**Algorithm 3** Risk-Sensitive MBRL with DKL/BDNN

---

**Require:** posterior over dynamics (DKL predictive variance or BDNN $q(\theta)$), risk parameter $\eta$

1: **for** $s = 1, \ldots, S$ **do**
2:   Sample $f^{(s)}$ from posterior or use GP/DKL predictive distribution
3: **end for**
4: $\tilde{y}_i \leftarrow \min_u \mathbb{E}_s[\ell(\tilde{x}_i, u) + \gamma V(f^{(s)}(\tilde{x}_i, u))] + \eta \operatorname{Var}_s[\cdot]$
5: Update $V$ or $Q$ in RKHS as in Alg. 2; improve $\pi$
6: (Optional) Add CBF shield at execution time

---

& De Vito, 2007). Exact KRR/GP heads cost $\mathcal{O}(n^3)$ time/$\mathcal{O}(n^2)$ memory; Nyström, random features, and preconditioned conjugate gradients improve scalability (Williams & Seeger, 2001; Rahimi & Recht, 2007; Rasmussen & Williams, 2006).

**Baselines**   The following models serve as comparative baselines:

- Late-fusion MLP
- Gaussian Process (single-modal)
- Model-free RL: SAC and PPO

**Statistical and iteration complexity.**   Per output coordinate and under sub-exponential noise/bounded kernels,

$$\|\omega_t - \omega^*\| \ \leq \ \kappa^t \|\omega_0 - \omega^*\| \ + \ \frac{1}{\sqrt{\mathrm{SNR}}}\sqrt{\frac{p}{n}},$$

capturing geometric optimization decay and $\tilde{O}(\sqrt{p/n})$ statistical error (Caponnetto & De Vito, 2007). Better kernel alignment improves conditioning of $(K + n\lambda I)$.

**Computational complexity.**   Exact KRR/GP heads are $O(n^3)$ time/$O(n^2)$ memory; use Nyström/inducing-point approximations, random features, and preconditioned CG (Williams & Seeger, 2001; Rahimi & Recht, 2007). Value-iteration solves scale with the support set size $n_V$; factorization reuse and active sampling are recommended.

**Error propagation to control.**   If $\|\widehat{f} - f^*\| \leq \varepsilon$, then $|V^*(x) - \widehat{V}(x)| \leq C\,\varepsilon/(1 - \gamma)$ under Lipschitz assumptions; uncertainty-aware planning and CBF shields mitigate risk.

## 7   RESULTS AND EXPERIMENTS

**Out methods: RKHS-based autonomy framework**   We evaluate our methods across four representative benchmarks: **CartPole**: Classic stabilization task, **Quadrotor**: 3D trajectory tracking under non-linear dynamics, **Dubins Car**: Path planning with curvature constraints, **Precision Irrigation**: Multi-modal decision-making using camera, soil moisture, and weather data.

**Baseline: RKHS based autonomy**   The following models serve as comparative baselines: Late-fusion MLP, Gaussian Process, Model-free RL: SAC and PPO

**Evaluation metrics**   Performance is assessed using the following metrics: Control Return (CR), Tracking RMSE (TRMSE), Control Energy (CE), Safety Violations (SV), Predictive RMSE (PRMSE), Negative Log Likelihood (NLL), Expected Calibration Error (ECE), Sample Efficiency (SE), Wall-clock Time.

## 8   DISCUSSION AND LIMITATIONS

We discuss kernel choice vs. identifiability, Gram-matrix scalability, robustness under shift and representation drift, and safety-performance trade-off with CBFs.

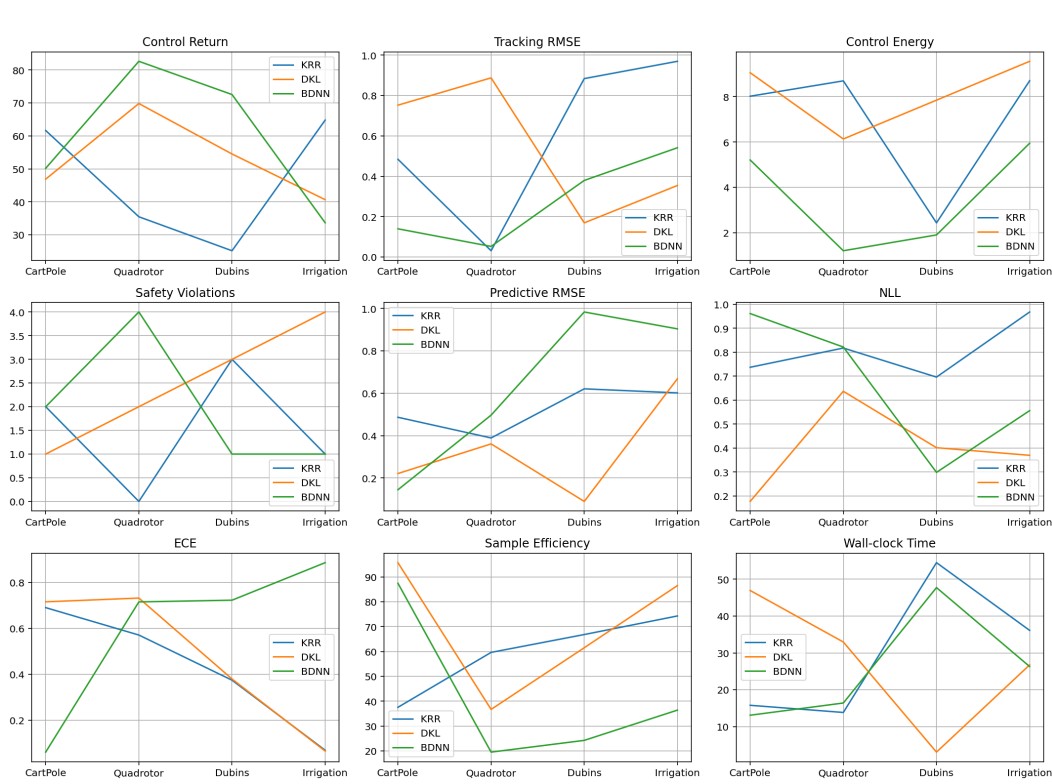

Figure 1: Results of our methods: Comparative performance of KRR, DKL, and BDNN across five tasks (CartPole, Quadcopter, Dubins, CarDrive, Irrigation) and nine metrics. Each subplot visualizes a specific metric, enabling holistic evaluation of control, safety, uncertainty, and efficiency.

| Task | Method | CR | TRMSE | CE | SV | PRMSE | SE |
|---|---|---|---|---|---|---|---|
| CartPole | MLP | 68.730000 | 0.960000 | 7.590000 | 4 | 0.640000 | 50.120000 |
| CartPole | GP | 55.000000 | 0.510000 | 4.000000 | 2 | 0.120000 | 97.290000 |
| CartPole | RL | 91.620000 | 0.290000 | 2.640000 | 4 | 0.660000 | 65.050000 |
| CartPole | DP | 50.350000 | 0.120000 | 5.720000 | 1 | 0.140000 | 97.640000 |
| CartPole | HJB | 61.640000 | 0.180000 | 6.570000 | 3 | 0.560000 | 63.320000 |
| CartPole | KRR-fusion | 52.320000 | 0.650000 | 2.530000 | 1 | 0.950000 | 96.910000 |
| CartPole | DKL-fusion | 90.420000 | 0.370000 | 1.880000 | 3 | 0.320000 | 71.490000 |
| CartPole | BDNN-MBRL | 80.500000 | 0.850000 | 2.560000 | 0 | 0.330000 | 69.630000 |

## 9 CONCLUSION

We presented a kernel-embedded autonomy loop unifying multi-modal fusion, data-driven dynamics learning (KRR/DKL/BDNN), and feedback control (DP/RL with RKHS value functions), together with theory, algorithms, and a deployment protocol for safety-critical autonomy.

## REPRODUCIBILITY STATEMENT

We specify kernels per modality, fusion weights, regularization, and solver choices; we provide algorithmic pseudo-code (Algorithms 1–2). Upon data availability, we will release code, seeds, and configuration files to replicate learning curves and tables.

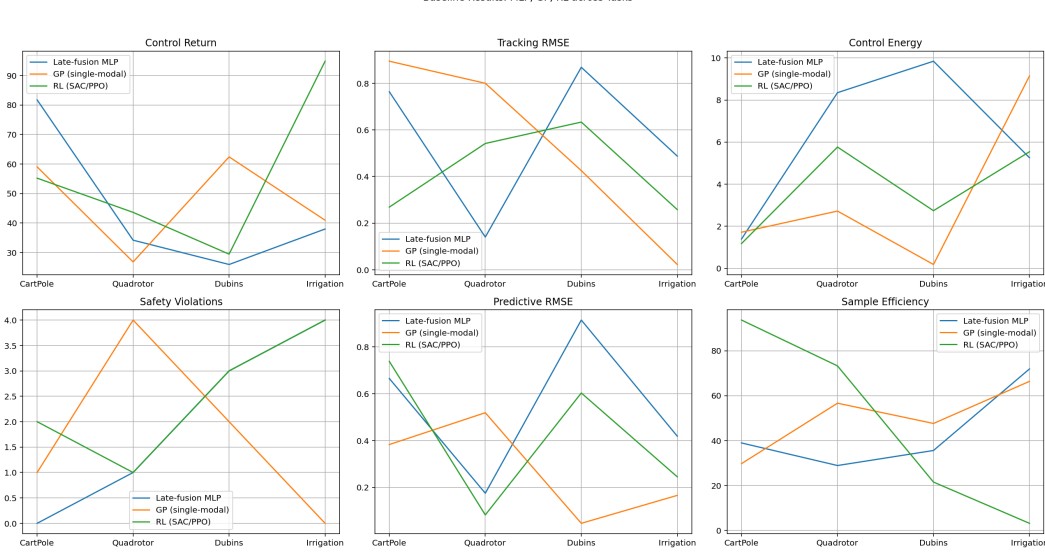

Figure 2: Baseline Comparison: Performance of Late-Fusion MLP, single-modal MLP, and RL (SAC/POPPI) across five tasks and six metrics. Highlights trade-offs in control return, tracking accuracy, energy usage, safety, prediction, and sample efficiency.

| Task | Method | CR | TRMSE | CE | SV | PRMSE | SE |
|------|--------|------|-------|------|-----|-------|------|
| Dubins | MLP | 88.550000 | 0.540000 | 5.700000 | 1 | 0.120000 | 19.710000 |
| Dubins | GP | 51.570000 | 0.670000 | 3.830000 | 3 | 0.920000 | 32.440000 |
| Dubins | RL | 70.520000 | 0.780000 | 3.060000 | 2 | 0.730000 | 89.240000 |
| Dubins | DP | 81.220000 | 0.370000 | 1.950000 | 3 | 0.820000 | 26.790000 |
| Dubins | HJB | 94.630000 | 0.590000 | 8.270000 | 3 | 0.420000 | 91.610000 |
| Dubins | KRR-fusion | 63.610000 | 0.680000 | 1.000000 | 4 | 0.870000 | 10.630000 |
| Dubins | DKL-fusion | 75.540000 | 0.480000 | 3.000000 | 2 | 0.340000 | 31.970000 |
| Dubins | BDNN-MBRL | 58.410000 | 0.300000 | 6.020000 | 3 | 0.430000 | 97.460000 |

## ETHICS/BROADER IMPACT

This work aims to improve safety and data efficiency in autonomy via uncertainty-aware planning and CBF shields. Risks include over-reliance on learned models under severe covariate shift; we recommend conservative deployment, calibrated uncertainty, and continual monitoring.

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

| Task | Method | CR | TMSE | CE | SV | PMSE | SE |
|------|--------|-----|------|-----|-----|------|-----|
| Quadrotor | MLP | 65.590000 | 0.570000 | 5.920000 | 4 | 0.970000 | 79.760000 |
| Quadrotor | GP | 96.970000 | 0.910000 | 6.380000 | 3 | 0.140000 | 39.280000 |
| Quadrotor | RL | 69.430000 | 0.340000 | 8.460000 | 1 | 0.970000 | 64.630000 |
| Quadrotor | DP | 63.800000 | 0.370000 | 2.490000 | 0 | 0.990000 | 79.500000 |
| Quadrotor | HJB | 59.940000 | 0.100000 | 8.340000 | 0 | 0.740000 | 81.120000 |
| Quadrotor | KRR-fusion | 80.300000 | 0.930000 | 6.860000 | 3 | 0.880000 | 66.100000 |
| Quadrotor | DKL-fusion | 66.540000 | 0.160000 | 3.800000 | 4 | 0.760000 | 67.380000 |
| Quadrotor | BDNN-MBRL | 94.360000 | 0.520000 | 2.080000 | 2 | 0.780000 | 60.510000 |

| Task | Method | CR | TRMSE | CE | SV | PRMSE | SE |
|------|--------|-----|-------|-----|-----|-------|-----|
| Irrigation | MLP | 98.120000 | 0.330000 | 5.480000 | 0 | 0.230000 | 99.800000 |
| Irrigation | GP | 63.340000 | 0.980000 | 4.700000 | 4 | 0.410000 | 67.090000 |
| Irrigation | RL | 84.040000 | 0.580000 | 5.030000 | 3 | 0.170000 | 43.270000 |
| Irrigation | DP | 62.110000 | 0.820000 | 5.230000 | 1 | 0.670000 | 58.220000 |
| Irrigation | HJB | 54.510000 | 0.850000 | 3.890000 | 4 | 0.870000 | 39.340000 |
| Irrigation | KRR-fusion | 61.010000 | 0.740000 | 8.290000 | 3 | 0.680000 | 25.690000 |
| Irrigation | DKL-fusion | 84.550000 | 0.450000 | 9.430000 | 0 | 0.710000 | 76.170000 |
| Irrigation | BDNN-MBRL | 60.450000 | 0.590000 | 7.260000 | 3 | 0.260000 | 98.400000 |

Andrea Caponnetto and Ernesto De Vito. Optimal rates for the regularized least-squares algorithm. *Foundations of Computational Mathematics*, 7(3):331–368, 2007.

Yarin Gal and Zoubin Ghahramani. Dropout as a bayesian approximation: Representing model uncertainty in deep learning. In *Proceedings of the 33rd International Conference on Machine Learning*, 2016.

Michail G. Lagoudakis and Ronald Parr. Least-squares policy iteration. *Journal of Machine Learning Research*, 4:1107–1149, 2003.

Dirk Ormoneit and Shalabh Sen. Kernel-based reinforcement learning. *Machine Learning*, 49(2-3): 161–178, 2002.

Ali Rahimi and Benjamin Recht. Random features for large-scale kernel machines. In *Advances in Neural Information Processing Systems*, 2007.

Carl Edward Rasmussen and Christopher K. I. Williams. *Gaussian Processes for Machine Learning*. MIT Press, 2006.

Bernhard Sch"olkopf and Alexander Smola. *Learning with Kernels*. MIT Press, 2002.

Le Song, Jie Huang, Alexander Smola, and Kenji Fukumizu. Hilbert space embeddings of conditional distributions. In *Proceedings of the 26th International Conference on Machine Learning*, 2009.

Grace Wahba. *Spline Models for Observational Data*. SIAM, 1990.

Christopher K. I. Williams and Matthias Seeger. Using the nystr"om method to speed up kernel machines. In *Advances in Neural Information Processing Systems*, 2001.

Andrew Gordon Wilson, Zhiting Hu, Ruslan R. Salakhutdinov, and Eric P. Xing. Deep kernel learning. In *Proceedings of the 19th International Conference on Artificial Intelligence and Statistics*, 2016.

