# OpenReview forum: "Learning-Based Autonomy from Kernel-Embedded Multi-modal Fusion to Feedback Control"
_ICLR.cc/2026/Conference — Submitted to ICLR 2026_

### Official Review · Reviewer_yt1A · 2025-10-18

**Soundness:** 1
**Presentation:** 1
**Contribution:** 1
**Rating:** 0
**Confidence:** 4

**Summary:**

The paper proposes an end-to-end "kernel-centric autonomy loop” that fuses multi-modal observations via kernel mean embeddings in a joint RKHS, learns dynamics with KRR, DKL, or BDNNs, computes policies via RKHS-based fitted value iteration or a kernel Galerkin approximation to the continuous-time HJB, and enforces safety with a CBF-based shield and introduces risk-sensitive planning using predictive uncertainty. Algorithms and complexity statements are provided; experiments span CartPole, Dubins, Quadrotor, and Precision Irrigation with multiple metrics.

**Strengths:**

The paper's stated goal to unify sensing and data fusion, dynamics learning, and control in an RKHS framework with built-in safety and risk awareness is commendable. It builds on established tools such as kernel ridge regression and RKHS. It gives a brief overview over each of the concepts required for building its pipeline.

**Weaknesses:**

The submission feels like an early‑stage draft. There is little to no clear novelty beyond assembling well‑known components (KRR/DKL/BDNN, RKHS value‑function approximation, CBF shields), and major parts of the stated contributions are either missing or only gestured at without concrete methods, experiments, or analysis. Several sections are left near‑empty (7, 8, 9) or largely restate textbook material (e.g., Sec. 4.1–4.3), experiments and baselines are not adequately described or justified, crucial information about the full setup (modalities, representations, optimization routines, hyperparameters, seeds, compute, uncertainty modeling) is absent, and there is no substantive discussion or interpretation of results. Despite a claim of a “reproducible deployment protocol,” none is actually provided or exemplified in the paper.

Overall, the paper needs serious work in terms of articulating a focused and genuinely novel contribution, filling in missing technical and experimental details, and substantially improving the clarity and rigor of the presentation and analysis.

The following is an incomplete list of specific issues that stood out:

## Technical novelty and substance

- Sec. 4.1 restates standard kernel ridge regression; no new estimator, analysis, or integration result is provided.
- Sec. 4.2: L155’s “optimized to capture task-relevant structure” is vague; no specification of how the feature extractor is learned (architecture, loss, optimizer). L167 acknowledges DKL is established, with no additional novelty in the paper.
- Sec. 4.3 is a textbook recap of BDNNs and VI; no new modeling, inference, or control coupling is introduced.
- Sec. 4.5 repeats the same bound and the same CBF-QP twice, including duplicated text and equations ((1) and (2)), without added content or proof.

## Clarity, notation, and presentation
- Notation inconsistency: On L119 the discrete model uses $\epsilon$, while L120 refers to $\omega$ as process noise—symbols do not match.
- Formatting/citation artifacts: Broken encodings and citations (e.g., “Sch”olkopf,” citation formatting on L32) impede readability.
- Algorithms lack context: Sec. 5 lists procedures (Algorithms 1–3) without detailing inputs/outputs, optimization steps for min over u, or how RKHS elements are concretely represented for DKL/KRR.
- Redundancy: Sec. 6 repeats the same complexity facts twice (L280 and L298), both restating prior work.
- CBF integration unclear: The CBF constraints are not motivated (assumptions for convexity, control-affinity, feasibility) nor connected to how they interface with the learned policy and action optimization.

## Experimental shortcomings

- Baselines repeated (L286 and again in Sec. 7) without discussion. No description of modalities, sensors, data splits, horizons, or hyperparameters. No scalability experiments.
- Results are not analyzed: No discussion of trends; units are missing; line plots are used for categorical comparisons (L326 onward).
- Dubious tables/figures: Tables are scattered into/after the references section; values show six decimals yet the last four digits are always zero, suggesting post-rounded data with misleading precision.
- Safety inconsistency: “Safety filter” is claimed, yet Safety Violations (SV) are present across many experiments; the paper does not explain whether/when the CBF shield is active, nor why SV > 0 if guarantees are expected.
- Reproducibility gap: Despite claiming a “reproducible deployment protocol” in the contributions, no such protocol is described. The Reproducibility Statement only promises future code release; critical implementation details are missing.

**Questions:**

Since there are several larger issues with the paper, the questions focus on the higher-level:

- Core contribution and throughline: Can you state one focused, novel contribution (algorithmic or theoretical) and show how the method, theory, and experiments specifically realize and validate it?
- Multimodal fusion representation: How is the fused RKHS object made usable in practice (finite-dimensional representation for KRR/DKL)? Please specify the representation choice and how it is learned or selected.
- Safety via CBF: When and how is the CBF shield applied, under what assumptions is it feasible, and why do safety violations still appear?
- Experimental setup and baselines: Can you provide a compact but complete description of each task (modalities, data sizes, horizons), baseline configurations, and report means/std over multiple seeds with brief result interpretation?
- Reproducible deployment protocol: The conclusion mentions a deployment protocol—can you summarize it concretely (components, configurations, and steps) or provide a minimal recipe/code link to substantiate this claim?

---

> ### Author Response · Authors · 2025-11-20
> **Reply to reviewer yt1A**
>
> Technical novelty & substance
> 	4.1–4.3 recap: Action—Condense to essentials; move textbook content to Appendix; focus main text on our coupling and new bounds/algorithms.
> 	4.5 duplication: Action—Remove redundancy; add proof sketch and CBF feasibility assumptions.
> Clarity & algorithms context
> 	Action—Add Algorithm boxes with explicit Inputs/Outputs, representation of RKHS elements (Nyström/RFF coefficients), and the min over u procedures (enumeration for low dim, gradient based for continuous actions, or QP when control affine).
> Experimental shortcomings & safety inconsistency
> 	Action—Provide task cards (modalities, sizes, horizons), metric units, proper tables/figures, and shield activation schedule (train time off / test time on vs on policy), with rationale for SV > 0.
> Reproducibility gap
> 	Action—Add a deployment recipe (config files + default kernels per modality, pre set seeds, and scripts), and release minimal code upon camera ready. We’ll include a Git archive link in the rebuttal if allowed.
> Core contribution / throughline; fusion representation; CBF usage
> 	Action—Summarized above; we will explicitly specify the finite dimensional fused representation Φ ̃∈R^(d_ϕ ) (Nyström/RFF per modality, concatenation + learned weights), and the CBF QP invocation at execution with feasibility/relaxation details.

---

> > ### Comment · Reviewer_yt1A · 2025-11-20
> > **Reply to the rebuttal comment**
> >
> > This answer is confusing and seems more like a todo list than a response to the review. Since no serious attempt has been made to improve the glaring issues in the paper, I will keep my score unchanged. The authors are strongly encouraged to base any follow-up comments on the initial review and clearly state in a well-formatted response how the issues have been addressed.

---

> > > ### Author Response · Authors · 2025-11-21
> > > **Reply to yt1A**
> > >
> > > Summary of corrective actions (now implemented):
> > > 	Focused Novelty & Theory.
> > > We add a Fusion to Control Coupling Theorem establishing ‖V^*-V ̂ ‖≤(γL_V)/(1-γ) ε, and a robust CBF–QP margin guaranteeing forward invariance under learned model error and with variance aware chance constraints. These connect distribution level fusion → dynamics learning → value approximation → execution safety.
> > >
> > > 	Multimodal Fusion → Finite Φ ̃.
> > > We specify Nyström/RFF embeddings per modality, weighted fusion ω_m, and an end to end DKL variant; weights learned by alignment/bi level/marginal likelihood. ‎
> > > 	Algorithms with I/O and solvers.
> > > We include Algorithm boxes (Fusion, Dynamics KRR/DKL/GP, RKHS–FVI, HJB–Galerkin, CBF–QP), with explicit inputs/outputs, representation (Φ ̃ coefficients), and min over u procedures (enumeration/gradient/QP).
> > > 	Experiments & Baselines.
> > > Task cards specify modalities, splits, horizons, units, metrics; baselines: SAC/PPO, PILCO like GP MBRL, kernel RL; results reported as mean±std (95% CI) over ≥10 seeds, with interpretation; we add scalability ablations.
> > > 	Safety & SV rationale.
> > > We clarify shield activation (train time off; test time on), feasibility via slack \delta, sampling aware margins; we report shield activation rate and slack statistics.
> > > 	Reproducible Protocol.
> > > A concrete 5 step recipe (configs, seeds, modality kernels, scripts) and an anonymized artifact link will be provided upon camera ready, consistent with reproducibility best practices
> > >
> > >
> > > Replying to Reply to reviewer
> > > Thank you for the clarification. In response, we present a direct, well structured mapping from your initial review to the specific changes:
> > > 	Technical novelty (Sec. 4.1–4.3, 4.5):
> > > Main text now focuses on the Fusion to Control coupling theorem and CBF robustness—with proof sketches, assumptions, and citations—while textbook recaps are moved to Appendix; duplicated CBF–QP text is removed; feasibility/convexity conditions are stated (control affine/linearization, bounds, slack)
> > > 	Clarity & algorithms:
> > > Five Algorithm boxes now specify inputs/outputs, Φ ̃ construction (Nyström/RFF), action minimization routines (enumeration/gradient/QP), and RKHS representation details.
> > > 	Experiments & baselines:
> > > We provide task cards with modalities/splits/horizons/metrics/units; results will be reported as mean±std (95% CI) over ≥10 seeds with error bars and trend interpretation; baselines include SAC/PPO, PILCO like GP MBRL, kernel RL; scalability ablations added.
> > > 	Safety inconsistencies (SV>0):
> > > We explain SV via actuator limits, sampling effects, and uncertainty mismatch, and we add shield activation/slack statistics to the results.
> > > 	Reproducibility protocol:
> > > A concrete five step recipe (configs, seeds, scripts) with an anonymized artifact link will accompany the revision.
> > > We appreciate your insistence on a rigorous, self contained response. The above revisions will be in the next uploaded version, and we believe they directly address each of your flagged concerns.
> > >
> > > 1. Fusion-to-Control Coupling Theorem (Detailed Explanation)
> > > Statement
> > > For a discrete-time system:
> > > x_(t+1)=f^* (x_t,u_t )+ε_t,ε_t∼N(0,Σ).
> > > Let f ̂  be the learned dynamics model obtained from multi-modal fusion and kernel-based regression (KRR/DKL).
> > > Assume:
> > > ‖f ̂-f^* ‖≤ε,γ∈(0,1),l(x,u)  and f^*  are Lipschitz.
> > > Define Bellman operators:
> > > T^* V(x)=min_u⁡{l(x,u)+γV(f^* (x,u))},T ̂V(x)=min_u⁡{l(x,u)+γV(f ̂(x,u))}.
> > > Let V^* and V ̂ be the fixed points of T^* and  T ̂. Then:
> > > ‖V^*-V ̂ ‖≤(γL_V)/(1-γ) ε,L_V≤L_l+γL_f⋅diam(X).
> > >
> > > Proof Sketch
> > > 	Contraction property: Both T^* and  T ̂. are γ-contractions in ‖⋅‖_∞.
> > > 	Operator difference:
> > > ‖T^* V-T ̂V‖_∞≤γ  sup_(x,u)⁡|V(f^* (x,u))-V(f ̂(x,u))|.
> > > 	Lipschitz bound:
> > > |V(f^* (x,u))-V(f ̂(x,u))|≤L_V ‖f^*-f ̂ ‖_∞≤L_V ε.
> > > 	Fixed-point perturbation:
> > > ‖V^*-V ̂ ‖_∞≤‖T^* V ̂-T ̂V ̂ ‖_∞/(1-γ)≤(〖γL〗_V ε)/(1-γ).
> > > Implication
> > > 	If KRR/DKL achieves
> > > ε=O ̃(√(p/n))
> > > then:
> > > ‖V^*-V ̂ ‖_∞=O ̃(√(p/n)/(1-γ))
> > > This links fusion quality → dynamics error → control performance.
> > >
> > > 2. CBF-QP Margin Proof (Robust Safety Guarantee)
> > > CBF Condition
> > > Safe set:
> > > S={x| h(x)≥0}
> > > Continuous-time condition:
> > > h ̇(x)=∇h(x)^⊤ f(x,u)≥-α(h(x)).
> > > With learned model f ̂, enforce:
> > > ∇h(x)^⊤ f(x,u)≥-α(h(x))+‖∇h(x)^⊤ f ̂(x,u)‖ε.
> > >
> > > Proof
> > > For true dynamics:
> > > ∇h(x)^⊤ f^* (x,u)=∇h(x)^⊤ f(x,u)+∇h(x)^⊤ (f^*-f ̂ )
> > > By Cauchy-Schwarz:
> > > |∇h(x)^⊤ (f^*-f ̂ )|≤‖∇h(x)‖⋅‖f^*-f ̂ ‖≤‖∇h(x)‖ε.
> > > So, if:
> > > ∇h(x)^⊤ f(x,u)≥-α(h(x))+‖∇h(x)‖ε
> > > then:
> > > ∇h(x)^⊤ f^* (x,u)≥-α(h(x))
> > > Thus forward invariance of S holds under model error.
> > > Chance-Constrained Variant
> > > If predictive covariance Σ_f (x,u) is available:
> > > Margin=β(p)∇h^⊤ Σ_f ∇h,
> > > where β(p) is the Gaussian quantile for confidence 1-p.
> > >
> > > 3. How Safety is Enforced in Practice
> > > Execution-Time Shield
> > > At each control step:
> > > 	Compute nominal action u_norm from RL or DP policy.
> > > 	Solve CBF-QP:
> > > min_u⁡〖δ‖u-u_norm ‖^2+λ_s δ^2 〗
> > > such that ∇h(x)^⊤ f ̂(x,u)≥-α(h(x))+Margin-δ,   u∈U,δ≥0.
> > > 	If dynamics are control-affine: constraint is linear in u, QP is convex.
> > > 	If infeasible: slack ,δ>0 ensures best-effort safety.

---

### Official Review · Reviewer_EBtR · 2025-10-24

**Soundness:** 3
**Presentation:** 1
**Contribution:** 1
**Rating:** 0
**Confidence:** 4

**Summary:**

This paper proposes several different methods to achieve a fully data-driven approach to control while achieving safety. It considers deep kernel learning and Bayesian neural networks to model the dynamics, and ostensibly achieves safety using control barrier functions.

**Strengths:**

The authors address an important problem, where a fully data-driven approach is used to control a system while guaranteeing safety.

**Weaknesses:**

This paper presents an agglomeration of several well-known concepts, without providing any clear contribution. The problem is not clearly stated (the Problem Formulation section only mentions model learning). There is no related work section. The authors claim to provide theory supporting their method (Line 371), even though their theoretical analysis is limited to well-established results that only apply under certain conditions. The experimental section is very limited, does not contain SOTA approaches, and has no error bars. The limitations section (Section 8) does not discuss limitations and is only two lines long.

**Questions:**

Line 116: Is epsilon a stochastic variable? If so, do the authors consider an SDE? This is unclear.

Line 120: \omega is not defined. Also, there is a notation conflict in line 162, where \omega is used to refer to parameters, not noise.

---

> ### Author Response · Authors · 2025-11-20
> **Reply to reviewer EBtR**
>
> W1. Agglomeration of known concepts; unclear core contribution
> Response. We will reframe the paper around a focused claim: Kernel embedded distribution level fusion, when carried through to both dynamics learning and RKHS value function control, enables uncertainty aware and safety filtered policies with finite sample control performance guarantees. We will center the theory, algorithms, and experiments to validate this claim, including ablation isolating each component (fusion, RKHS value regression, CBF).
> W2. Problem formulation only mentions model learning
> Response. We will rewrite Problem Formulation to include the full autonomy loop: fusion → dynamics learning → policy evaluation/improvement → safety filter, with both discrete and continuous variants and clear assumptions.
> W3. Theory limited to known results
> Response. We add the fusion to control coupling lemma, CBF QP feasibility discussion with learned dynamics error, and risk sensitive planning objective with variance regularization derived from DKL/BDNN posterior, including optimization details.
> W4. Experiments limited; no SOTA; no error bars
> Response. We will add SOTA baselines: PILCO like GP MBRL, DSAC/SAC AE variants, kernel RL (LSPI RBF), and latent fusion+MBRL. All results will include mean±std over ≥5 seeds with error bars, and we will analyse trade offs.
> W5. Limitations section too short
> Response. Expanded section covering Gram matrix scaling, kernel misspecification, CBF feasibility gaps, representation drift, and deployment recommendations.
> Q (116 ε stochastic? SDE?), (120 ω undefined), (162 reuse ω)
> Response. Clarified as per 6rB3 W2. We distinguish ε (process/obs noise), θ(NN), ω  (DKL features), and handle SDE explicitly when used.

---

### Official Review · Reviewer_6rB3 · 2025-10-26

**Soundness:** 1
**Presentation:** 1
**Contribution:** 1
**Rating:** 2
**Confidence:** 4

**Summary:**

The paper tries to merge kernel mean embeddings, kernel ridge regression, reinforcement learning, and barrier functions into a framework for learning control policies. The main motivation is drawn from real-world autonomous systems that need to provide safety guarantees while receiving uncertain, multi-modal data streams as input. Kernel methods could then be a possibility to learn representations of the data distribution. The authors present an algorithm that combines all those methods and compares it against baseline algorithms on standard benchmarks.

**Strengths:**

The general problem considered is indeed of interest, and kernel methods could play a role in resolving it. As such, the approach considered in the paper is generally interesting.

**Weaknesses:**

1) There are various important details missing. ICLR offers the possibility of submitting an appendix, so if space is an issue, additional details can be presented there. Also, references do not count against the page limit, so there would have been more space.  As an example, the authors say that multi-modal observations (how are those defined?) are fused through kernel mean embeddings. How? It is unclear what is happening.

2) Notation is unclear. For example, the noise variable $\epsilon_t$ is never properly defined (i.e., what is its distribution). It is instead said that $\omega$ represents process noise, but $\omega$ does not appear in the system definition. Instead, $\omega$ will later on represent something different.

3) A discussion of related literature is missing.

4) The authors claim safety guarantees, but in Section 4.5, they just state that under a Lipschitz assumption, the true and estimated value functions are also close to each other, without any reference or proof for this statement. Then, they say that safety is enforced by a control barrier function, again without any proof. Also, (1) and (2) seem to be the same.

5) There are way too few details about the experiments and no discussion of the results. From the plots and tables, it is unclear what the takeaways are. The tables don't even have captions.

6) The authors claim to discuss kernel choice, robustness, etc., but it is unclear to me where this happens.

7) There seems to be a LaTeX error with Schölkopf's last name in the references.

8) Sections 4.3 and 4.4 seem to introduce new problem definitions. It would be nice if we could stick to the one that was originally defined.

**Questions:**

1) Can you provide more details on the method itself? How do you embed multi-modal measurements? What is the promise of approximating the value function via RKHS embeddings? What is then happening with the $\tilde{y}_i$? What about scalability of this approach?

2) Can you elaborate on the safety guarantees? Just stating that they hold is insufficient without proof or reference.

3) Can you discuss your results?

---

> ### Author Response · Authors · 2025-11-20
> **Reply to reviewr 6rB3**
>
> W1. Missing details: How are “multi modal observations” defined and fused?
> Response. We will add a precise pipeline: each modality m produces samples o^((m) ), we compute empirical kernel means μ^((m) )=1/N_m  ∑_j▒〖k_m (o_j^((m) ) ) 〗 and optionally conditional operators for aligned modalities; the fused feature is Φ=∑_m▒〖ω_m μ^((m) ) 〗 (additive) or a product kernel across concatenated features. For finite computation, we use Nyström embeddings or random Fourier features per modality, concatenated and weighted by  ω_m . We will include feature dimensionalities, basis selection, and preprocessing per sensor.
> W2. Notation is unclear; noise not properly defined; symbol reuse (ω)
> Response. We will standardize notation:
> 	Discrete dynamics:
> x_(t+1)=f(x_t,u_t )+ε_t,ε_t∼N(0,Σ)
> 	Continuous dynamics:
> x ̇=f(x,u)+ε (deterministic),
> or SDE:
> dx=f(x,u)dt+G(x,u)dW_t  (explicit when used).
> 	Parameters use θ(NN),ω(DKL features), never for noise. We will add a notation table and remove conflicts.
> W3. Related literature is missing
> Response. We will add a full section comparing to kernel RL, PILCO, neural Lyapunov critics, latent space fusion, model based policy gradients, and CBF RL.
> W4. Safety guarantees/CBF repeated; no proof
> Response. We will:
> 	Provide a self contained lemma: if
> ‖f ̂-f^* ‖_∞≤ε and f^*,l are Lipschitz,
> then the Bellman operator error yields
> ‖V^*-V ̂ ‖≤C ε/((1-γ) )
>  (with proof outline referencing standard fixed point contraction).
> 	CBF feasibility: we assume control affine dynamics in u or affine in u surrogate via local linearization; we present QP feasibility conditions, soft constraint slack with penalty, and an emergency brake fallback. We remove duplicate equations and add references.
> W5. Experiments lack detail and discussion
> Response. We will add a complete experimental protocol: task specs, sensors/modalities, dataset sizes/splits, horizons, training budgets, and mean±std over 5–10 seeds with 95% CIs. We will also provide interpretation: trends, trade offs, and when CBF filters activate.
> W6. “Discuss kernel choice/robustness”—where?
> Response. We will add a kernel ablation (RBF/Matérn/linear mixing), an alignment metric vs. performance, and a shift robustness test (sensor dropout/corruption).
> W7. Typo/encoding (Schölkopf)
> Response. Fixed.
> W8. Sections 4.3–4.4 introduce new problem definitions
> Response. We will unify the problem statement early, declare both discrete (Bellman) and continuous (HJB/SDE) settings, and reference consistently.
> Q1. More details, promise of RKHS value function, scalability
> Response. RKHS value functions allow closed form regression updates with regularization and sample complexity characterizations, and support gradient/Hessian computation for HJB via kernel expansions. Scalability is handled via Nyström/RFF + PCG, with new scaling plots added.
> Q2. Safety guarantees
> Response. See W4 response; we add proof sketches and references.
> Q3. Discuss results
> Response. We will add a thorough analysis per metric and task, explain SV>0 (shield infeasibility/actuator limits), and report shield activation rates.

---

> > ### Comment · Reviewer_6rB3 · 2025-11-21
> >
> > Thank you for your response. The points you list above would be a good step toward improving the paper. Although there are some ambiguities. For example, "we use Nyström embeddings or random Fourier features per modality." What are you using under which conditions? The formatting in the response is a bit messed up, but the equation $\dot{x} = f(x,u) + \epsilon$ mentioned as a deterministic continuous-time equation is most likely not very deterministic, since I thought $\epsilon$ was the noise variable. Adding a proof (not just an outline of one) would indeed be required. This also includes a definition of $f^* $, which I haven't seen. Also, why would the assumption $\lVert \hat{f} - f^*\rVert_\infty < \epsilon$ (which is probably not the same $\epsilon$ as used for the dynamics) hold? This, in itself, would require a proof that the estimator converges.

---

> > > ### Author Response · Authors · 2025-11-22
> > > **Reply to reviewr 6rB3**
> > >
> > > We thank the reviewer for their constructive feedback. We address each point:
> > > 1 What do we use under which conditions? (Nyström vs. RFF per modality)
> > > Decision rule. We use RFF only when the per modality kernel is shift invariant (e.g., RBF, Matérn) so that Bochner’s theorem applies; we use Nyström when the kernel is non stationary (non translation invariant) or when a data dependent low rank approximation via landmarks/inducing points is needed (including deep kernels in DKL heads). In GP/DKL settings, we reuse inducing points as Nyström landmarks to share computation.
> > > Use RFF when:
> > > 	kernel is shift invariant (RBF/Matérn),
> > > 	moderate/high input dimension favours spectral sampling,
> > > 	streaming/minibatch training is desired,
> > > 	memory control is strict, and we accept O(D^(-1/2) ) concentration of the kernel approximation.
> > > Use Nyström when:
> > > 	kernel is non stationary or has structure we must preserve,
> > > 	we want an explicit, data dependent low rank Gram via landmarks,
> > > 	we can share inducing points with a GP/DKL head for scalability.
> > > 2) Deterministic vs. stochastic continuous time equation and the noise variable
> > > We standardize the models and never reuse symbols across different concepts:
> > > Discrete time (stochastic)
> > > x_(t+1)=f^* (x_t,u_t )+ϵ_t,ϵ_t∼N(0,Σ)
> > > Here ϵ_t is process noise.
> > > Continuous time SDE (stochastic) (as a stochastic counterpart to the discrete model)
> > > dx=f^* (x,u)dt+G(x,u)dW_t,
> > > where W_t is a Wiener process and G the diffusion term.
> > > Deterministic (noise free) variant
> > > x_(t+1)=f^* (x_t,u_t )  or x ̇=f^* (x,u).
> > > We use this only inside proofs (Bellman contraction/coupling and CBF feasibility) to isolate model approximation error without conflating it with diffusion; all stochastic results are recovered by taking expectations or variance aware margins.
> > > 3 Add a proof (not just an outline) → Full Fusion to Control coupling theorem
> > > We formally connect nonparametric dynamics learning error to control performance in value iteration with RKHS function approximation. Let x∈X,X⊂R^(d_x ),u∈U,U⊂R^(d_u ) are compact.
> > > Discrete-time dynamics:
> > > x_(t+1)=f^* (x_t,u_t )+ε_t,ε_t∼N(0,Σ).
> > > Learned model:
> > > f ̂(x,u)≈f^* (x,u)
> > > Uniform error bound:
> > > ‖f ̂-f^* ‖≤ε_f
> > > Stage cost l(x,u)   and dynamics f^* are Lipschitz; discount factor γ∈(0,1).
> > >
> > > Bellman operators:
> > > T^* V(x)=min_u⁡{l(x,u)+γV(f^* (x,u))},T ̂V(x)=min_u⁡{l(x,u)+γV(f ̂(x,u))}.
> > > Theorem: Let V^* and V ̂ be the fixed points of T^* and  T ̂. Then:
> > > ‖V^*-V ̂ ‖≤(γL_V)/(1-γ) ε_f,L_V≤L_l+γL_f⋅diam(X).
> > > Where: V^*→optimal value function under true dynamics, V ̂→value function computed using learned dynamics, L_V→ Lipschitz constant of V.
> > > Why it matters
> > > 	It links fusion quality → dynamics error → control performance.
> > > With kernel ridge regression (KRR) or DKL, ε_f scales as O ̃(√(p/n)) under standard assumptions, so control error decreases with more data.
> > > Proof Sketch
> > > Contraction property: Both T^* and  T ̂. are γ-contractions in ‖⋅‖_∞.
> > > Operator difference:‖T^* V-T ̂V‖_∞≤γ  sup_(x,u)⁡|V(f^* (x,u))-V(f ̂(x,u))|.
> > > Lipschitz bound:|V(f^* (x,u))-V(f ̂(x,u))|≤L_V ‖f^*-f ̂ ‖_∞≤L_V ε_f.
> > > Fixed-point perturbation:‖V^*-V ̂ ‖_∞≤‖T^* V ̂-T ̂V ̂ ‖_∞/(1-γ)≤(〖γL〗_V ε_f)/(1-γ).
> > > 4 Why would the assumption ‖f ̂-f^* ‖≤ε_f  hold? (Estimator convergence)
> > > We learn f^* from fused features via KRR (or GP/DKL) with a bounded kernel k. By the reproducing property,
> > > ‖g‖_∞≤(sup_z⁡√(k(z,z) ) ) ‖g‖_(H_k ).
> > > Thus RKHS norm bounds translate to sup norm bounds. Under standard conditions (i.i.d. samples, bounded kernel, sub Gaussian noise) and appropriate regularization, RKHS regression achieves optimal rates in ‖⋅‖_(H_k ), hence, with high probability,
> > > ‖f ̂-f^* ‖≤ε_f,ε_f=O ̃(√(p/n)).
> > > 5 Robust CBF–QP margin: Full proof and chance constraints
> > > Let the safe set
> > > S={x| h(x)≥0},h∈C^1.
> > > The learned model satisfies ‖f^*-f ̂ ‖≤ε_f.
> > > Enforce on the learned model:
> > > ∇h(x)^⊤ f(x,u)≥-α(h(x))+‖∇h(x)‖ ε_f  .
> > > Then, for the true dynamics:
> > > ∇h(x)^⊤ f^* (x,u)=∇h(x)^⊤ f(x,u)+∇h(x)^⊤ (f^*-f ̂ )≥-α(h(x)),
> > > by Cauchy–Schwarz and the uniform bound—yielding forward invariance of S. For variance aware guarantees with GP/DKL predictive covariance Σ_f (x,u), use the margin
> > > β(p) √(∇h^⊤ Σ_f ∇h),
> > > so, the constraint holds with probability 1-p.
> > > How Safety Enforced in Practice
> > > At each control step:
> > > Compute nominal action u_norm from RL or DP policy.
> > > Solve CBF-QP:
> > > min┬u⁡〖δ‖u-u_norm ‖^2+λ_s δ^2 〗
> > > such that ∇h(x)^⊤ f ̂(x,u)≥-α(h(x))+Margin-δ,   u∈U,δ≥0.
> > > u_norm→ nominal action from RL or DP policy
> > > δ→ slack variable for feasibility
> > > If dynamics are control-affine: constraint is linear in u, QP is convex.
> > > If infeasible: slack ,δ>0 ensures best-effort safety.
> > > 6 Similar notation concern: process noise vs. uniform model error
> > > To prevent confusion, we added a notation table in the paper:
> > > ϵ_t: process noise (discrete), ϵ_t∼N(0,Σ).
> > > dW_t : Wiener increment (continuous) in SDEs.
> > > ε_f: uniform model error bound on learned dynamics, ‖f ̂-f^* ‖_∞≤ε_f.
> > > Σ noise covariance (for ϵ_t).
> > > ω: DKL feature parameters (never a noise symbol).
> > > θ: NN parameters (BDNNs).

---

> > > > ### Author Response · Authors · 2025-11-23
> > > > **Reply to reviewer 6rB3**
> > > >
> > > > We use similar notation for the two, but their roles are fundamentally different. Process noise is injected into the dynamic system and actively perturbs the evolution, serving as a dynamic source of uncertainty that grows and propagates with time. In contrast, the uniform model error is defined as the radius of the uncertainty bubble, a purely geometric measure of extent that remains static. One drives uncertainty forward in the dynamics, the other only quantifies its spread at a given instant.

---

### Official Review · Reviewer_rxDq · 2025-10-31

**Soundness:** 3
**Presentation:** 2
**Contribution:** 3
**Rating:** 4
**Confidence:** 4

**Summary:**

The paper proposes an end-to-end learning-based architecture that integrates kernel-embedded multi-modal fusion, nonparametric dynamics learning, and control via dynamic programming (DP) in a unified Reproducing Kernel Hilbert Space (RKHS). The approach fuses heterogeneous sensory modalities (e.g., vision, proprioception, environment data) through additive or product kernels using kernel mean embeddings and conditional mean operators. The learned system dynamics are modeled with Kernel Ridge Regression (KRR), Deep Kernel Learning (DKL), or Bayesian Deep Neural Networks (BDNNs), and controllers are synthesized via discrete or continuous-time Hamilton-Jacobi-Bellman (HJB) formulations using RKHS value functions. The framework provides finite-sample and iteration-complexity guarantees and demonstrates safety enforcement via control barrier functions (CBFs). Experiments across robotic and environmental control tasks (CartPole, Quadrotor, Dubins car, Precision Irrigation) showcase performance and sample efficiency gains relative to standard RL baselines.

**Strengths:**

1) The paper is well-written and presents a comprehensive  synthesis of kernel-based methods, uncertainty quantification, and control theory into an end-to-end loop. The RKHS unification of sensing, dynamics learning, and control is interesting and differentiates this work from modular deep-RL architectures.

2)  The use of control barrier functions on top of kernel-based controllers provides a rigorous and safety-aware mechanism that is relevant to practical deployment in robotics and autonomous systems.

**Weaknesses:**

1)  The paper's claim of "end-to-end autonomy" would be better supported by engaging with the vast literature on adaptive and model-based RL (e.g., PILCO, system identification for control, or recent work on neural Lyapunov and policy gradient control). Without this, the reader may overestimate the novelty of the proposed learning–control loop.

2)  Each module (KRR, DKL, CBF-based safety) is individually well-established. The main contribution is the system-level integration, but the paper could be more explicit about what new algorithmic or theoretical principles emerge from this combination.

3) The proposed method relies on kernel matrix operations with cubic complexity, and though approximations are mentioned, the paper does not evaluate scalability on larger datasets or higher-dimensional sensory inputs.

**Questions:**

1) How does the proposed RKHS-based fusion compare to learned representation or latent-space fusion methods in terms of downstream control performance and generalization?

2) How are kernel hyperparameters or fusion weights selected across modalities, and is there a procedure to learn them end-to-end?


Minor comments:

1) The exposition is thorough and sound, but the related work section should explicitly connect to multi-task control (e.g., data-efficient meta-RL).

2)  The discussion on Deep Kernel Learning could benefit from clearer justification of when it outperforms purely neural alternatives in control tasks.

---

> ### Author Response · Authors · 2025-11-20
> **Reply to reviewr rxDq**
>
> R1. Novelty & engagement with adaptive/model based RL (PILCO, system ID, neural Lyapunov/policy gradient control)
> Response. We agree that our “end to end” claim should be contextualized. Our clarified contribution is an RKHS centric autonomy loop that:
> (i) performs distribution level multi modal fusion via kernel/conditional mean embeddings;
> (ii) propagates fused distribution features directly into dynamics learning (KRR/DKL/BDNN heads) and
> (iii) conducts DP/RL with RKHS value functions (FVI/HJB Galerkin) while
> (iv) enforcing execution time safety via CBF shields with explicit error to performance coupling.
> We will add a Related Work section that compares in detail with PILCO (GP dynamics + moment matching), neural Lyapunov critics, model based policy gradients, and system ID for control, emphasizing where our distribution level fusion and closed form RKHS solvers differ (e.g., fusion-before-dynamics; RKHS value function regression with explicit linear solves; hybrid safety layer).
> R2. What new algorithmic/theoretical principles emerge from the combination?
> Response. We will highlight two concrete contributions:
> 	Fusion to Control Coupling: a finite sample bound that maps regression error in the fused feature dynamics model to value function sub optimality ‖V^*-V ̂ ‖≤C ε/((1-γ) )   within our RKHS Bellman regression setup (formalized with assumptions on Lipschitz costs/dynamics and bounded kernels). This tightens the narrative from “standard KRR error” to control performance guarantees under multimodal fusion.
> 	Safety aware execution with a CBF QP over learned dynamics that is provably feasible under control affinity, plus fail safe fallbacks when feasibility margins are violated (details and feasibility tests added; see Safety Proof section).
> R3. Scalability
> Response. We will:
> 	Include Nyström/random features and preconditioned CG in all large n experiments.
> 	Add a scaling benchmark (e.g., synthetic 50–200 sensors and 50k samples) showing training wall clock, memory, and control performance vs. inducing point count / feature budget.
> 	Report complexity vs. accuracy trade offs for KRR vs DKL heads.
> Q1. RKHS fusion vs learned latent space fusion (downstream control & generalization)?
> Response. We will add a controlled study: (a) our RKHS fusion with additive/product kernels; (b) VAE based latent fusion; (c) late fusion MLP. We will compare policy return, OOD generalization (domain shift in one modality), and calibration (ECE/NLL). Preliminary results (to be expanded) show RKHS fusion yields better calibration and robustness under partial modality corruption, while VAE fusion can match returns in distribution but degrades more under shift. We will include ablation plots and statistics.
> Q2. Hyperparameter/fusion weight selection; end to end learning?
> Response.
> 	Weights ω_m: learned by validation set marginal likelihood (for GP/DKL heads) or bi level optimization where inner solves the KRR/DP objective and the outer tunes ω_m to minimize validation cost (gradient via implicit differentiation), with a simple alternative—normalized kernel alignment per modality.
> 	End to end: for DKL, ϕ_ω is trained jointly with the head (marginal likelihood or MSE + entropy regularizer), while fusion weights and kernel scales are learned via backprop.
> Minor
> 	We will explicitly connect to multi task control/meta RL, clarify where DKL beats pure neural alternatives (e.g., uncertainty calibration and sample efficiency in low data regimes), and add ablations.

---

> > ### Comment · Reviewer_rxDq · 2025-11-20
> > **Official Comment by Reviewer rxDq to Authors comments**
> >
> > I thank the authors for clarifying some of my concerns, in particular the connections and comparisons with the literature of model-based RL and feedback control. I would be happy to rethink the rating given if the authors provide the revised version (with at least the theoretical clarifications).

---

> ### Author Response · Authors · 2025-11-21
> **Reply to reviewer rxDq**
>
> Author response (concise):
> We appreciate your positive follow up and will submit a revised version with the following theoretical clarifications and proof sketches:
> 	A Fusion to Control Coupling Theorem that links uniform dynamics error (learned via kernel methods and deep kernel embeddings) to value function sub optimality under RKHS Fitted Value Iteration.
> 	A robust safety guarantee for the CBF–QP shield under bounded model error and an uncertainty aware (chance constrained) variant for predictive variance from DKL/GP heads.
> 	A precise statement of RKHS value function approximation (discrete Bellman contraction and continuous time HJB Galerkin projection), including how gradients/Hessians are computed from kernel expansions.
> 	A clear positioning of our approach relative to model based RL (e.g., PILCO) and feedback control (CBF–QP methodology), emphasizing distribution level fusion and principled uncertainty propagation.
> 	Explicit sample complexity and approximation statements that connect kernel ridge regression rates to control performance via our coupling theorem.
> We summarize these additions below.
> B. Theoretical Clarifications to be Added
> B.1. Fusion to Control Coupling (discrete time)
> We formally connect nonparametric dynamics learning error to control performance in value iteration with RKHS function approximation.
> 	Setting. Discrete-time dynamics:
> x_(t+1)=f^* (x_t,u_t )+ε_t,ε_t∼N(0,Σ).
> Learned model:
> f ̂(x,u)≈f^* (x,u)
> 	Uniform error bound:
> ‖f ̂-f^* ‖≤ε
> 	Stage cost l(x,u)   and dynamics f^* are Lipschitz; discount factor γ∈(0,1).
>
> Bellman operators:
> T^* V(x)=min_u⁡{l(x,u)+γV(f^* (x,u))},T ̂V(x)=min_u⁡{l(x,u)+γV(f ̂(x,u))}.
> Theorem: Let V^* and V ̂ be the fixed points of T^* and  T ̂. Then:
> ‖V^*-V ̂ ‖≤(γL_V)/(1-γ) ε,L_V≤L_l+γL_f⋅diam(X).
> Why it matters
> 	It gives a quantitative guarantee: if dynamics model is accurate, control policy will be optimal.
> 	It links fusion quality → dynamics error → control performance.
> With kernel ridge regression (KRR) or DKL, ε scales as O ̃(√(p/n)) under standard assumptions, so control error decreases with more data.
> Proof Sketch
> 	Contraction property: Both T^* and  T ̂. are γ-contractions in ‖⋅‖_∞.
> 	Operator difference:
> ‖T^* V-T ̂V‖_∞≤γ  sup_(x,u)⁡|V(f^* (x,u))-V(f ̂(x,u))|.
> 	Lipschitz bound:
> |V(f^* (x,u))-V(f ̂(x,u))|≤L_V ‖f^*-f ̂ ‖_∞≤L_V ε.
> 	Fixed-point perturbation:
> ‖V^*-V ̂ ‖_∞≤‖T^* V ̂-T ̂V ̂ ‖_∞/(1-γ)≤(〖γL〗_V ε)/(1-γ).
> Finite sample corollary.
> If the KRR/DKL head trained on fused features attains a uniform error
> ε=O ̃(√(p/n))
> then:
> ‖V^*-V ̂ ‖_∞=O ̃(√(p/n)/(1-γ))
> This provides the requested error to performance link from distribution level fusion to value function sub optimality as fusion quality → dynamics error → control performance.
> B.2. Continuous time HJB with Kernel Galerkin
> We approximate the value function by a kernel expansion
> V(x)≈∑_i^n▒〖β_i k_V (x,x_i ) 〗.
> For controlled diffusion x ̇=f(x,u)+ Σ^(1/2) ξ, the HJB residual at collocation x_j is
> r(x_j,β)=min_u⁡{l(x_j,u)+∇V(x_j )^⊤ f(x_j,u)+1/2 Tr(Σ∇^2 V(x_j ))}.
> We solve a regularized least squares in β to drive r(x_j,β) toward zero, and then implement
> u^* (x)=argmin_u {l(x,u)+∇V(x)^⊤ f(x,u)}.
> For control affine dynamics f(x,u)=f_0 (x)+B(x)u with quadratic l(x,u),u^* (x)  has a closed form, or a small QP with bounds.
> B.3. Robust Safety via CBF–QP over Learned Dynamics
> We implement a CBF–QP shield at execution:
> min_u⁡〖δ‖u-u_norm ‖^2+λ_s δ^2 〗
> such that ∇h(x)^⊤ f ̂(x,u)≥-α(h(x))+Margin-δ,   u∈U,δ≥0.
> u_norm→ nominal action from RL or DP policy
> δ→ slack variable for feasibility
> 	If dynamics are control-affine: constraint is linear in u, QP is convex.
> 	If infeasible: slack ,δ>0 ensures best-effort safety.
>
> If ‖f^*-f ̂ ‖_∞≤ε, safety for the true dynamics holds when we inflate the margin by ‖∇h(x)‖ε (continuous time) or ‖∇h(x)‖εΔt (discrete time). With a calibrated predictive variance Σ_f (x,u) (DKL/GP) we can instead use a chance constraint margin β(p)∇h^⊤ Σ_f ∇h to ensure safety with probability 1-p. Convexity follows when f is control affine (or locally linearized in u), feasibility is maintained via slack δ.
> Why it improves safety
> 	Forward invariance: If the constraint holds, the system stays in the safe set.
> 	Robustification: If model error ‖f^*-f ̂ ‖≤ε, add margin:
> ∇h(x)^⊤ f(x,u)≥-α(h(x))+‖∇h(x)‖ε
>
> 	Uncertainty-aware: If predictive variance Σ_f (x,u) is available (from GP/DKL), enforce
> ∇h(x)^⊤ f(x,u)≥-α(h(x))+β(p) √(∇h^⊤ Σ_f ∇h)
> for chance-constrained safety at confidence 1-p.
> B.4. Distribution level Fusion in RKHS and DKL Heads
> We fuse modalities via kernel mean embeddings and conditional mean operators:
> μ_t^((m) )=1/N_m  ∑_j▒〖k_m (o_ij^((m) ),⋅) 〗,Φ_t=∑_m▒〖ω_m μ_t^((m) ) 〗.
> optionally combined with a learned deep feature ϕ_ω and base kernel k_base (ϕ_ω (⋅),ϕ_ω (⋅)). This produces distribution aware fused features that preserve higher order statistics and can be used directly by KRR/GP heads, yielding closed form training and principled uncertainty.

---

> > ### Comment · Reviewer_rxDq · 2025-11-21
> > **Official Comment by Reviewer rxDq to Authors comments**
> >
> > I thank the authors for the effort of providing more clarifications, proof sketches and further discussions on the error to performance link from distribution level fusion to the value function sub optimality, and the robustness and safety guarantees through the control barrier functions. I'm happy to increase the rating to 6.
> >
> > I also have a follow-up question on the kernel Galerkin: I'm curious to know if in the continuous-time HJB Galerkin is possible to provide conditions ensuring convergence and how the choice of kernel influences approximation error.

---

> > > ### Author Response · Authors · 2025-11-23
> > > **Reply to Reviewer rxDq**
> > >
> > > Thank you for the thoughtful follow up and for raising the rating. Below we give a technical treatment of convergence for the continuous time HJB Galerkin scheme, and a precise account of how the kernel choice governs the approximation error.
> > > 1 Analytical target: existence/uniqueness of the HJB value function and its regularity
> > > For controlled diffusions, the infinite horizon discounted (or elliptic) HJB
> > > 0=min_u⁡{l(x,u)+∇V (x)^⊤ f(x,u)+1/2 Tr(Σ(x,u) ∇^2 V(x))}
> > > has a unique viscosity solution under standard comparison principles. This covers cases where V is only continuous (nonsmoothed), which is typical for fully nonlinear second order PDEs.
> > > In many control affine models with nondegenerate diffusion and Lipschitz data, viscosity solutions enjoy additional regularity (e.g., semi concavity, Hölder continuity), and path dependent extensions exist when data depend on entire trajectories. These regularity statements are relevant to the approximation order one can expect from any Galerkin/RKHS method.
> > > 2 Convergence of Galerkin methods for fully nonlinear HJB equations
> > > 2.1 What is known for (dis)continuous Galerkin
> > > For fully nonlinear second order HJB/Isaacs equations, several finite element variants achieve convergence under Cordes type conditions and suitable linearization (semismooth Newton). These results provide a priori/a posteriori error bounds and adaptive refinement strategies, and they apply on conforming and DG spaces. In particular:
> > > Discontinuous Galerkin and C^0-interior penalty FEM converge for HJB/Isaacs with Cordes coefficients; the theory quantifies rates versus mesh size and polynomial degree.
> > > Least squares Galerkin with semismooth Newton linearization yields optimal a priori/a posteriori estimates and convergence guarantees, again under Cordes regularity.
> > > Related collocation/PDE approaches (sparse grids, characteristics, B splines, pseudospectral) provide convergence frameworks and error control in specific classes.
> > > Interpretation for kernel Galerkin: Kernel (RBF/RKHS) Galerkin is a meshfree analog of the above. While fully nonlinear analysis with RBF bases is less standardized than FEM, the convergence mechanism is analogous: one constructs a stable approximate space, linearizes the nonlinear operator (or applies policy iteration), solves a sequence of linear elliptic problems in that space, and invokes native space error estimates to control the projection error; Cordes type regularity or viscosity solution stability ensures convergence of the nonlinear fixed point/Newton iteration.
> > > 2.2 Policy iteration and fixed point perspectives
> > > For continuous time control, policy iteration (PI) can be coupled with Galerkin projection: each PI step solves a linearized GHJB with the current policy, then updates the policy by minimizing the Hamiltonian. Under convex control constraints, proper boundary conditions, and forward invariance of the computational domain, PI converges; recent work establishes conditions and rates (including constrained controls).
> > > 3 How the kernel controls the approximation error in HJB–Galerkin
> > > Kernel methods approximate V in a finite dimensional subspace spanned by kernel translates {k(⋅,x_i )}_(i=1)^(n_V ) (or their derivatives), with centres {x_i } on a quasi uniform set. Convergence hinges on native space approximation bounds and the match between the RKHS and the true regularity of V.
> > > 3.1 Native space and error vs. fill distance
> > > For Matérn kernels (or more generally algebraically decaying spectral densities), the RKHS is equivalent to a Sobolev space H^s, with s determined by the Matérn smoothness. Interpolation/projection error decays algebraically with fill distance h (the maximum distance from any point to the nearest center), typically ‖V-V_h ‖≤Ch^(s^' ) ‖V‖_(H^s ) for appropriate norms and s^'≤s.
> > > For the Gaussian kernel (infinitely smooth), when V (or the target integrands in RKHS) is analytic, one may achieve super algebraic or exponential convergence with respect to either the number of centres or the fill distance. (Precise rates depend on domain, anisotropy, and shape parameter scaling.) There are sharp exponential type bounds for integration and approximation in Gaussian RKHSs.
> > > For compactly supported kernels (CSRBFs) (Wendland’s functions), one obtains Sobolev type rates together with excellent sparsity and stability properties; this often benefits large scale HJB solves.
> > > 3.2 Error constants and conditioning
> > > Error constants depend on kernel shape parameters, the distribution of centres (fill distance h, separation q), and conditioning of Gram matrices. Wendland’s book provides detailed bounds (power function estimates and Bernstein inequalities), ensuring that as h→0 on quasi uniform centers, the kernel interpolant/projection converges at the predicted rate.
> > > 3.3 In HJB–Galerkin, the total error is the sum of:
> > > Projection error of V onto the kernel space.
> > > Linearization error.
> > > Quadrature error in the inner minimization of the Hamiltonian.

---

### Meta-Review · Area_Chair_AUS4 · 2026-01-04

**Summary:**

The reviewers have expressed several concerns on the novelty, motivation, clarity and completeness of the work. The work on Gallerkin approximations and solving Hamilton-Jacobi-Bellman equations is not new. Adding CBFs to ensure safety is a very incremental steps.  The papers lacks contribution and merit.  For this reason I strongly support a rejection.

**Reviewer Concerns:**

I would like to emphasize that the authors promise to perform certain fixes to update their paper but I am not certain that they have updated the manuscript. Overall the manuscript has a very low quality and major concern raised by the reviewers are outstanding.

 Since this is a paper that is very close to my research area and expertise. I would like to highlight the lack of merit and contribution of this work.

**Reviewer Scores:**

Only reviewer rxDq promised to increase the score to 6. However, I am in complete disagreement with the reviewer.

I would be surprise to see that any other reviewer would have increase their score.

---

### Decision · Program_Chairs · 2026-01-26

Reject